# Exploring the chemical composition, in vitro and in silico study of the anticandidal properties of annonaceae species essential oils from the Amazon

**Márcia Moraes Cascaes[1]\*, Silvia Helena Marques da Silva[2], Mozaniel Santana de Oliveira[3,4]\*, Jorddy Neves Cruz[5], Ângelo Antônio Barbosa de Moraes[5], Lidiane Diniz do Nascimento[3], Oberdan Oliveira Ferreira[3], Giselle Maria Skelding Pinheiro Guilhon[1], Eloisa Helena de Aguiar Andrade[1,3]**

**1** Programa de Pós-Graduação em Química, Universidade Federal do Pará, Belém, PA, Brazil, **2** Seção de Bacteriologia e Micologia LabMicol—SABMI Laboratório de Micologia, Instituto Evandro Chagas—IEC/SVS/MS, Ananindeua, Brazil, **3** Laboratório Adolpho Ducke, Coordenação de Botânica, Museu Paraense Emílio Goeldi, Belém, Brazil, **4** Programa de Pós-Graduação em Ciências Biológicas—Botânica Tropical, Universidade Federal Rural da Amazônia and Museu Paraense Emílio Goeldi, Belém, PA, Brazil, **5** Laboratory of Functional and Structural Biology, Institute of Biological Sciences, Universidade Federal do Pará, Belém, PA, Brazil

\* cascaesmm@gmail.com (MMC); mozaniel.oliveira@yahoo.com.br (MSO)

**Data Availability Statement:** All relevant data are within the paper.

## Abstract

Chemical composition of the essential oils (EOs) from the leaves of five *Annonaceae* species found in the amazon region was analyzed by Gas chromatography coupled to mass spectrometry. The antifungal activity of theses EOs was tested against *Candida albicans*, *Candida auris*, *Candida famata*, *Candida krusei* and *Candida tropicalis*. In addition, an in silico study of the molecular interactions was performed using molecular modeling approaches. Spathulenol (29.88%), α-pinene (15.73%), germacra-4(15),5,10(14)-trien-1-α-ol (6.65%), and caryophylene oxide (6.28%) where the major constitents from the EO of *Anaxagorea dolichocarpa*. The EO of *Duguetia echinophora* was characterized by β-phellanderene (24.55%), cryptone (12.43%), spathulenol (12.30%), and sabinene (7.54%). The major compounds of the EO of *Guatteria scandens* where β-pinene (46.71%), α-pinene (9.14%), bicyclogermacrene (9.33%), and *E*-caryophyllene (8.98%). The EO of *Xylopia frutescens* was characterized by α-pinene (40.12%) and β-pinene (36.46%). Spathulenol (13.8%), *allo*-aromadendrene epoxide (8.99%), thujopsan-2-α-ol (7.74%), and muurola-4,10(14)-dien-1-β-ol (7.14%) were the main chemical constituents reported in *Xylopia emarginata* EO. All EOs were active against the strains tested and the lowest inhibitory concentrations were observed for the EOs of *D. echinophora*, *X. emarginata*, and *X. frutescens* against *C. famata* the Minimum Inhibitory Concentration values of 0.07, 0.019 and 0.62 μL.mL$^{-1}$, respectively. The fungicidal action was based on results of minimum fungicidal concentration and showed that the EOs showed fungicide activity against *C. tropicalis* (2.5 μL.mL$^{-1}$), *C. krusei* (2.5 μL.mL$^{-1}$) and *C. auris* (5 μL.mL$^{-1}$), respectively. The computer simulation results indicated that the major compounds of the EOs can interact with molecular targets of *Candida* spp.

**Funding:** The author M.M.C. thanks CAPES for the Phd scholarship process number: [88887.497476/2020-00]. The author A.A.B.d.M. thanks CNPq for the scientific initiation scholarship. The author Mozaniel Santana de Oliveira thanks PDPG-POSDOC - Programa de Desenvolvi-mento da Pós-Graduação (PDPG) Pós-Doutorado Estratégico, as well as CAPES for the scholars-hip (process number: (88887.852405/2023-00). The authors thanks Edital 02/2023 - PAPQ/PROPESP - Universidade Federal do Para. The funders had no role in study design, data collection and analysis, decision to publish, or preparation of the manuscript.

**Competing interests:** The authors have declared that no competing interests exist.

## 1. Introduction

Fungal infections pose a threat to global health and annually, more than 150 million serious cases of fungal infections in humans occur worldwide, resulting in approximately 1.7 million deaths a year [1, 2]. The number of fungal species resistant to currently marketed drugs is increasing [3]. Among the microorganisms that cause fungal infections, the most aggravating problems include the resistance of *Candida* species, which cause infections in humans, mainly *C. albicans* [4].

Although several synthetics drugs are commercially available for the control of infections caused by fungi, there is a need to explore new efficient antifungal drugs, which may include naturally occurring compounds such as essential oils (EOs). The antimicrobial action of the different types of essential oils against microorganisms has been subject of several studies [1, 2, 5, 6], and molecular targets that are targeted by commercial drugs for this disease include Sterol 14α-Demethylase Cytochrome P450 proteins [7].

The EOs are responsible for the characteristic smell and/or flavor of plants and come from their secondary metabolism [8]. From a chemical point of view EOs are organic and volatile substances, composed mostly of terpenoids and phenylpropanoids and generally have a molecular weight of less than 300 Da [6]. Among the constituents present in EOs, terpenoids represent a large group of phytochemicals with promising antimicrobial activity [9]. The interactions between the chemical constituents present in an EO can cause additive, antagonistic or synergistic effects. When the overall effect of the mixture of EO constituents is greater than the sum of the individual actions of these substances, a synergistic effect occurs, while the additive effect results from the combinations of the sum of the effect of each chemical substance. Antagonism is considered when the efficacy of two or more compounds is less than that of one of them [6].

EOs are produced by more than 17,500 species of plants belonging to several families of Angiosperms [8], such as *Annonaceae*. This family is recognized for the economic importance of the fruits, ethnobotany, raw material for cosmetic products, in addition to produce volatile chemical constituents with antimicrobial activity [10, 11]. Some EOs obtained from *Annonaceae* species showed relevant results against fungal microorganisms, such as *Anaxagorea brevipes* Benth. [12], *Annona vepretorum* Mart. [13], and *Duguetia lanceolata* A.St.-Hil [14].

Considering the increase in the incidence of fungal infections in humans, the limitations of available therapeutic strategies and the resistance to certain synthetic drugs, EOs represent promising natural sources for the development of effective antifungal drugs. In this study, the chemical composition and the antifungal activity of EOs from the leaves of some species of *Annonaceae* (*Anaxagorea dolichocarpa* Sprague & Sandwith, *Duguetia echinophora* R.E.Fr., *Guatteria scandens* Diels, *Xylopia emarginata* Mart. and *Xylopia frutescens* Aubl.) were evaluated against *Candida* spp. In addition, were used molecular modeling approaches to investigate how the main compounds of the EOs of these species can interact with Sterol 14α-Demethylase Cytochrome P450 proteins, molecular targets of *Candida* spp.

## 2. Materials and methods

### 2.1. Botanical material

The leaves of *A. dolichocarpa*, *D. echinophora*, *G. scandens*, *X. emarginata* and *X. frutescens* were collected in the municipality of Magalhães Barata (State of Pará, Amazon region, Brazil) in May 2019 (00˚47'51.6" S; 047˚33'38.4" W). The samples were identified by Dr. Jorge Oliveira, a parataxonomist from the Museu Paraense Emílio Goeldi (MPEG), Belém, Pará, Brazil. Voucher specimens were deposited at the Herbarium of MPEG under registration code MG-

237513 for *A. dolichocarpa*, MG-237493 for *D. echinophora*, MG-237508 for *G. scandens*, MG-237506 for *X. emarginata* and MG-237492 for *X. frutescens*.

## 2.2. Preparation of botanical material and extraction of essential oils

The leaves of *Annonaceae* species were dried in an air-circulation oven for five days at 35°C, after that they were crushed in a knife mill (Tecnal, model TE-631/3, Brazil). Moisture content was analyzed using a moisture analyzer (Marte, model ID50, Brazil). The EOs were extracted by hydrodistillation in a glass modified Clevenger-type apparatus [15], using 150 g of plant material for each experiment. Hydrodistillations were carried out for 3 h at 100°C. The EOs were dried over anhydrous sodium sulfate and stored in a freezer at -10°C. The yields of EOs (%) were calculated from the plant dry weight and expressed in mL/100 g of dried material.

## 2.3. Chemical composition analysis

The chemical composition of the EOs were analyzed using chromatography/mass spectrometry (GC/MS) using a Shimadzu QP Plus 2010 GC-MS (Kyoto, Japan) protocols reported earlier by our research group [16, 17]. The retention index was calculated for all volatile constituents using a homologous series of *n*-alkanes ($C_8$-$C_{40}$, Sigma-Aldrich, United States) according Van den Dool and Kratz [18], and the compounds were identified by comparing their mass spectrum and retention index with the data from the libraries [19].

## 2.4. Antifungal activity of the essential oils

**2.4.1. Microorganisms.** Five fungal species were used to analyze the antifungal activity and includes *Candida albicans* INCQS40175, *C. tropicalis* ATCC 6258, *C. famata* ATCC 62894, *C. krusei* ATCC 13803 and *C. auris* IEC-01. Each microorganism was cultivated in Sabouraud agar at 37°C for 48 h before the beginning of the tests. These tests were performed at the Laboratory of Superficial and Systemic Mycoses of Instituto Evandro Chagas (IEC), Ananindeua, Pará, Brazil.

**2.4.2. Agar disc diffusion test.** The antifungal activity of the EOs was evaluated using the agar disc diffusion method [20] with modifications. Paper filter discs (6 mm) were impregnated with 20 μL of each EO. The suspensions of the test microorganisms were prepared with 0.45% saline solution (0.5 on the McFarland scale). Each microorganism suspension was spread on the surface of Sabouraud agar culture medium in Petri dishes (15 × 90 mm). Then, paper discs impregnated with the extracts were placed on the surface of the plates inoculated with the microorganisms. The plates were incubated for 48 h at 37°C. After this period, visual readings were taken, observing the presence of growth inhibition zone measured in millimeters, with the help of a millimeter ruler. As a positive control, paper discs were impregnated with a 30 μL nystatin solution. All tests were carried out in duplicate. Inhibition zones ≥8 mm indicated that the microorganism was sensitive to the tested essential oil, according to the classification early proposed [21–23].

**2.4.3. Broth microdilution: Determination of MIC.** The susceptibility of the microorganisms to the Eos was determined by the broth microdilution method recommended by the "US National Committee for Clinical Laboratory Standards" (NCCLS) [24], with adaptations. The microorganisms were cultivated in Sabouraud agar at 37°C for 48 h. From these cultures, cellular suspensions similar to the McFarland scale 0.5 were prepared. In a 96-well plate, serial dilution at a ratio of 2 of each EO to be tested was performed, starting from 10% (20/180 μL), in a final volume of 100 μL. Then, 100 μL of the yeast suspension was added. The final concentration in each well reached 50 μL.mL$^{-1}$, 25 μL.mL$^{-1}$, 12.5 μL.mL$^{-1}$, 6.2 μL.mL$^{-1}$, 3.1 μL.mL$^{-1}$, 1.5 μL.mL$^{-1}$, 0.7 μL.mL$^{-1}$, 0.3 μL.mL$^{-1}$, 0.19 μL.mL$^{-1}$ and 0.0 μL.mL$^{-1}$. DMSO was used for the

dilutions. After adding the yeast to the previously diluted oils, the plate was incubated at 37°C for 48 h, and at the end, the readings were taken visually, and the lowest concentration of EO capable of inhibiting visible fungal growth was recorded. The test was carried out in duplicate [25].

**2.4.4. Broth microdilution: Determination of MFC.** After completion of the test and visual reading of the broth microdilution to determine minimum inhibitory concentration (MIC), the test to determine the minimum fungicidal concentration (MFC) was carried out. The test consisted of plating 10 μL of each dilution in Sabouraud agar and incubating at 37°C for 48 h. After this period, the lowest dilution capable of killing 99.5% of the original inoculum was recorded. The test was carried out in duplicate [25]. To calculate the minimum inhibitory concentrations (MIC), the protocols of Kowalska-Krochmal et al., [26] and minimum fungicidal concentration (MFC) Rex et al., [27], Both using Excel software.

## 2.5. Molecular docking

The molecular structure of β-phellandrene, cryptone, spathulenol, and pinene were obtained in PubChem; then their structures were optimized with B3LYP/6-31G* [28, 29] using Gaussian09 program [30]. We used the molecular docking method to evaluate the compound's interaction mode with Sterol 14α-Demethylase Cytochrome P450 (CYP51). For this, we used the Molegro Virtual Docker 5.5 [31, 32], and the crystal structure used as molecular targets can be found in the Protein Data Bank using the ID: 5TZ1 [7]. The MolDock Score (GRID) scoring function was used with a Grid resolution of 0.30 Å and 5 Å radius encompassing the entire connection cavity. The MolDock SE algorithm was used with a number of runs equal to 10, 1500 max interactions, and a max population size equal to 50. The maximum evaluation of 300 steps with a neighbor distance factor equal to 1 and an energy threshold equal to 100 was used during the molecular docking simulation.

# 3. Results and discussions

## 3.1. Chemical composition and yield of the essential oils

The chemical composition and the yield of the EOs obtained from *Annonaceae* species are reported in Table 1. The EO yields showed a variation between 0.16 and 1.53% and the highest yield was found for the EO from *X. frutescens* (1.53%).

In another study the yield found for the *X. frutescens* EO was 1.00% [33]. The EO of a specimen of *X. emarginata* collected in the Caxiuanã National Forest, Melgaço, State of Pará State, Brazil showed a higher yield (0.3%) than that described in the present work [34]. The yield in EOs from the fruits of *A. dolichocarpa* was 0.50% [35], while the EO yield in the leaves and fine stems yielded 0.20% and 0.10%, respectively [36].

Comparison between results found in this work with others library indicates that the variation in EO yields of *Annonaceae* species can be correlated with abiotic factors. In general, the EO yield of a plant species varies according to the part, seasonality and geographic distribution, among other factors [37, 38]. Understanding the factors that determine the EO yield of a plant species, especially those of commercial interest, is very important to optimize conditions that can improve it.

In total, 98 chemical constituents were identified in the five analyzed EOs, representing an average of 91.97%. Hydrocarbon monoterpenes were predominant in the EOs of *D. echinophora* (47.24%), *G. scandens* (57.15%) and *X. frutescens* (89.18%), while oxygenated sesquiterpenes were found in greater proportion in the EOs of *A. dolichocarpa* (51.15%) and *X. emarginata* (57.20%).

**Table 1.  Yield, and chemical composition of the essential oils of *Annonaceae* species.**

| | | | $A_{dol}$ | $D_{ech}$ | $G_{sca}$ | $X_{ema}$ | $X_{fru}$ |
|---|---|---|---|---|---|---|---|
| **Essential Oil Yield (%)** | | | **0.16** | **0.82** | **0.24** | **0.17** | **1.53** |
| $RI_L$ | $RI_C$ | Constituents (%) | | | | | |
| 924 | 920 | α-Thujene | | 0.18 | | | 6.05 |
| 932 | 929 | α-Pinene | 15.73 | 4.52 | 9.14 | | 40.12 |
| 969 | 960 | Sabinene | | 7.54 | | | |
| 974 | 976 | β-Pinene | | 5.13 | 46.71 | | 36.46 |
| 988 | 985 | Myrcene | | 2.03 | | | 0.77 |
| 1003 | 1000 | *p*-Mentha-1(7),8-diene | | 1.02 | | | |
| 1002 | 1000 | α-Phellandrene | | | | | 0.64 |
| 1014 | 1008 | α-Terpinene | | 0.37 | | | 0.19 |
| 1022 | 1012 | *o*-Cymene | | | | | 0.30 |
| 1025 | 1019 | Sylvestrene | | | 1.30 | | 4.26 |
| 1025 | 1022 | β-Phellandrene | | 24.55 | | | |
| 1054 | 1052 | γ-Terpinene | | 1.03 | | | 0.39 |
| 1086 | 1114 | Terpinolene | | 0.87 | | | |
| 1095 | 1093 | Linalool | | | 0.45 | | 0.66 |
| 1102 | 1100 | Perillene | | 0.12 | | | |
| 1135 | 1133 | *trans*-Pinocarveol | | 0.33 | | 1.38 | 0.35 |
| 1136 | 1134 | *trans*-*p*-Menth-2-en-1-ol | | 0.21 | | | |
| 1140 | 1135 | *trans*-Verbenol | 0.27 | | | | |
| 1137 | 1136 | *cis*-Verbenol | | | | | 0.12 |
| 1160 | 1156 | Pinocarvone | 0.48 | | | 0.42 | 0.15 |
| 1166 | 1162 | *p*-Mentha-1,5-dien-8-ol | 1.00 | | | 0.46 | |
| 1174 | 1171 | Terpinen-4-ol | | 1.9 | 0.15 | | 0.29 |
| 1183 | 1180 | Cryptone | | 12.43 | | | |
| 1186 | 1185 | α-Terpineol | | | 0.12 | 0.13 | 0.29 |
| 1194 | 1191 | Myrtenol | 2.05 | 0.23 | | 2.07 | 0.55 |
| 1204 | 1206 | Verbenone | | | | 0.66 | |
| 1207 | 1220 | *trans*-Piperitol | | 0.12 | | | |
| 1215 | 1214 | *trans*-Carveol | | 0.17 | | 0.11 | |
| 1224 | 1249 | 3-isopropyl-Phenol | | 2.13 | | | |
| 1227 | 1303 | Phellandral | | 2.54 | | | |
| 1239 | 1251 | Carvone | | 0.30 | | | |
| 1249 | 1248 | Geraniol | | | | | 0.13 |
| 1289 | 1319 | *p*-Cymen-7-ol | | 1.05 | | | |
| 1290 | 1285 | γ-Terpinen-7-al | | 0.32 | | | |
| 1335 | 1333 | δ-Elemene | | | 3.39 | 0.43 | 1.41 |
| 1348 | 1346 | α-Cubebene | | 0.10 | 0.17 | 0.11 | |
| 1369 | 1365 | Cyclosativene | | | | 0.24 | |
| 1373 | 1369 | α-Ylangene | | | | 0.35 | |
| 1374 | 1373 | α-Copaene | 2.50 | 0.85 | 2.48 | 0.30 | |
| 1374 | 1374 | Isoledene | | | 0.39 | | |
| 1379 | 1377 | Geranyl acetate | | | | | 0.10 |
| 1387 | 1382 | β-Bourbonene | | | | 0.19 | |
| 1389 | 1388 | β-Elemene | 2.86 | 2.43 | 0.65 | 1.62 | 0.18 |
| 1417 | 1438 | *E*-Caryophyllene | 0.64 | 1.33 | 8.98 | | |
| 1434 | 1431 | γ-Elemene | | | 2.11 | 0.41 | |

*(Continued)*

**Table 1.** (*Continued*)

| | | | $A_{dol}$ | $D_{ech}$ | $G_{sca}$ | $X_{ema}$ | $X_{fru}$ |
|---|---|---|---|---|---|---|---|
| **Essential Oil Yield (%)** | | | **0.16** | **0.82** | **0.24** | **0.17** | **1.53** |
| 1439 | 1437 | Aromadendrene | | | | 0.16 | |
| 1452 | 1452 | α-Humulene | 0.46 | 0.26 | 0.52 | 0.13 | |
| 1478 | 1478 | γ-Muurolene | | | | 2.16 | |
| 1484 | 1478 | Germacrene D | | | 1.75 | 1.30 | 0.39 |
| 1489 | 1488 | β-Selinene | | | 0.47 | 1.41 | |
| 1492 | 1491 | *cis*-β-Guaiene | | | | 0.61 | |
| 1493 | 1496 | *epi*-Cubebol | | | | 1.05 | |
| 1495 | 1499 | γ-Amorphene | | 0.20 | | | |
| 1500 | 1499 | α-Muurolene | | | | 0.89 | |
| 1500 | 1401 | Bicyclogermacrene | 1.37 | 0.19 | 9.33 | | 4.22 |
| 1500 | 1503 | α-Muurolene | 0.56 | 1.47 | | | |
| 1505 | 1506 | β-Bisabolene | | 0.28 | | | |
| 1505 | 1507 | *E,E*-α-Farnesene | | | | 0.26 | |
| 1509 | 1508 | α-Bulnesene | | 0.17 | | | |
| 1511 | 1510 | δ-Amorphene | | | 0.47 | | |
| 1513 | 1511 | γ-Cadinene | | | | 1.67 | |
| 1514 | 1512 | Cubebol | 1.67 | | | | |
| 1522 | 1519 | δ-Cadinene | 0.69 | 3.28 | | 3.57 | |
| 1532 | 1531 | γ-Cuprenene | | | | 0.2 | |
| 1544 | 1546 | α-Calacorene | | 0.88 | | 2.37 | |
| 1559 | 1588 | Germacrene B | | | 0.47 | | |
| 1562 | 1554 | *epi*-Longipinanol | | 0.14 | | | |
| 1564 | 1561 | β-Calacorene | | 0.15 | | | |
| 1577 | 1575 | Spathulenol | 29.88 | 12.30 | 5.26 | 13.80 | 0.81 |
| 1582 | 1580 | Caryophyllene oxide | 6.27 | 1.96 | 0.99 | | |
| 1586 | 1573 | Thujopsan-2-α-ol | | | | 7.74 | |
| 1590 | 1581 | Globulol | | | | | 0.14 |
| 1592 | 1593 | Viridiflorol | | | 0.19 | | |
| 1594 | 1591 | Salvial-4(14)-en-1-one | | | | 6.45 | |
| 1596 | 1592 | Fokienol | 2.01 | | | | |
| 1600 | 1606 | Rosifoliol | | | 0.19 | | |
| 1608 | 1618 | Humulene epoxide II | 2.89 | 0.64 | | 4.88 | |
| 1618 | 1646 | 1,10-di-*epi*-Cubenol | | | 0.15 | | |
| 1627 | 1633 | 1-*epi*-Cubenol | | 0.36 | | | |
| 1630 | 1631 | Muurola-4,10(14)-dien-1-β-ol | | | | 7.14 | |
| 1639 | 1645 | *allo*-Aromadendrene epoxide | 2.07 | | | 8.99 | |
| 1644 | 1635 | α-Muurolol (= Torreyol) | | | 0.76 | | |
| 1645 | 1648 | Cubenol | | | | 0.71 | |
| 1648 | 1650 | *cis*-Guaia-3,9-dien-11-ol | | | | | 0.22 |
| 1652 | 1658 | Himachalol | 0.30 | | | | |
| 1652 | 1670 | α-Cadinol | | 0.27 | | | |
| 1658 | 1672 | *neo*-Intermedeol | | | 1.75 | | |
| 1668 | 1671 | 14-hydroxy-9-*epi-E*-Caryophyllene | | 0.21 | | | |
| 1675 | 1681 | Cadalene | | | | 1.29 | |
| 1676 | 1682 | Mustakone | | | | 1.48 | |
| 1679 | 1676 | Khusinol | | | | 1.78 | |

(*Continued*)

**Table 1.** (Continued)

|  |  |  | A_{dol} | D_{ech} | G_{sca} | X_{ema} | X_{fru} |
|---|---|---|---|---|---|---|---|
| **Essential Oil Yield (%)** |  |  | **0.16** | **0.82** | **0.24** | **0.17** | **1.53** |
| 1685 | 1689 | Germacra-4(15),5,10(14)-trien-1-α-ol | 6.65 | 0.16 |  | 0.84 |  |
| 1690 | 1690 | Z-α-trans-Bergamotol | 2.41 |  |  |  |  |
| 1759 | 1753 | Benzyl benzoate |  |  | 0.44 |  |  |
| 1762 | 1770 | β-Acoradienol |  |  |  |  |  |
| 1765 | 1772 | β-Costol |  |  |  | 1.79 |  |
| 1767 | 1777 | 14-oxy-α-Muurolene |  |  |  | 0.55 |  |
| 2042 | 2042 | Kaurene |  |  |  | 0.32 |  |
| Hydrocarbon monoterpenes |  |  | 15.73 | 47.24 | 57.15 | - | 89.18 |
| Oxygenated monoterpenes |  |  | 3.80 | 21.85 | 0.72 | 5.23 | 2.54 |
| Hydrocarbon sesquiterpenes |  |  | 9.08 | 11.59 | 31.18 | 19.67 | 6.20 |
| Oxygenated sesquiterpenes |  |  | 51.15 | 16.04 | 9.73 | 57.20 | 1.27 |
| Others |  |  | - | - | - | 0.32 | - |
| Total |  |  | 82.76 | 96.72 | 98.78 | 82.42 | 99.19 |

$RI_C$ = Calculated retention index; $RI_L$ = Literature retention index; $A_{dol}$ = *Anaxagorea dolichocarpa*; $D_{ech}$ = *Duguetia echinophora*; $G_{sca}$ = *Guatteria scandens*; $X_{ema}$ = *Xylopia emarginata*; $X_{fru}$ = *Xylopia frutescens*.

Spathulenol (29.88%), α-pinene (15.73%), germacra-4(15),5,10(14)-trien-1-α-ol (6.65%) and caryophylene oxide (6.28%) where the major compounds of EO of *A. dolichocarpa*. The chemical composition described in this work was similar to that described by Andrade and coworkers, in which spathulenol (26.2%), α-pinene (16.8%) and β-pinene (12.3%) were described as the major constituents from the leaves EO of a especime of *A. dolichocarpa* collected in the same locality [36]. The hydrocarbon monoterpene α-pinene showed significant antifungal activity against *Candida parapsilosis* [39] and *C. albicans* [40]. Antifungal activity of caryophyllene oxide was reported in a previous study [41].

The EO of *D. echinophora* was characterized by β-phellanderene (24.55%), cryptone (12.43%), spathulenol (12.30%) and sabinene (7.54%). Although there are no studies reporting the chemical composition from the EOs of *D. echinophora*, the EOs others species belonging to the genus *Duguetia* were rich on α-pinene, β-pinene, spathulenol, viridiflorene, bicyclogermacrene, caryophyllene oxide, germacrene D, β-caryophyllene and humulene epoxide II [42].

The chemical composition of the EOs from *G. scandens* also being reported for the first time in the literature. This study showed that the monoterpenes β-pinene (46.71%) and α-pinene (9.14%) and the sesquiterpenes bicyclogermacrene (9.33%) and E-caryophyllene (8.98%) were the major constituents of the EO of this species. Compounds β-pinene, α-pinene and E-caryophyllene have already been reported as the main constituents of *G. hispida* [10], while bicyclogermacrene was described as one of the major constituents of *G. pogonopus* Martius [43] and *G. australis* A.St.-Hi [44]. The positive enantiomers of β-pinene and α-pinene have antimicrobial activity against *C. albican* [45], and E-caryophyllene shohwed antimicrobial activity against dermatophytic fungi [46]. Antimicrobial activity was also related to EOs that have bicyclogermacrene as major compounds [47].

Oxygenated sesquitepenes spathulenol (13.8%), *allo*-aromadendrene epoxide (8.99%), thujopsan-2-α-ol (7.74%) and muurola-4,10(14)-dien-1-β-ol (7.14%) were the main chemical constituents from the EO of *X. emarginata*. The EO from a specimen of *X. emarginata* collected in the Caxiuanã National Forest, Melgaço, State of Pará, Brazil, was characterized by high percentage of sesquiterpene spathulenol (73.00%) [34]. While the specimen colected in

Campo Grande, State of Mato Grosso do Sul, Brazil, showed spathulenol (34.40%), caryophyllene oxide (25.00%), myrtenal (7.7%) and *trans*-pinocarveol (6.3%) as the the major components [48].

The EO of *X. frutescens* was charactezed by the monoterpenes α-pinene (40.12%) and β-pinene (36.46%). Previous studies with EO from leaves of a specimen of *X. frutescens* collected in the Municipality of Capela, State of Sergipe, Brazil, presented as major compounds the sesquiterpenes (*E*)-caryophyllene (31.48%), bicyclogermacrene (15.13%), germacrene D (9.66%), δ-cadinene (5.44%), viridiflorene (5.09%) and α-copaene (4.35%) [33]. The EO of another specimen collected in the State of Paraíba, Brazil, was characterized by caryophyllene (23.91%), γ-cadinene (12.48%), β-ocimene (8.19%), cadin-4-en-10-ol (5.78%), δ-cadinene (5.7%), viridiflorol (4.83%) and γ-elemene (4.55%) [49]. While the sesquiterpenes bicyclogermacrene (23.23%), germacrene D (21.16%), (*E*)-caryophyllene (17.24%) and β-elemene (6.35%) were the major constituents of EO of a specimen collected in the Serra de Itabaiana National Park, in the City of Itabaiana, State of Sergipe, Brazil [50].

Although there are qualitative and quantitative differences, the EOs of the five species described in this work were characterized by compounds belonging to the class of mono and sesquiterpenes and this composition is in accordance to other *Annonaceae* species [11, 51, 52].

### 3.2. Antifungal activity

The antifungal activity of the EOs from the leaves of *Annonaceae* species was estimated in terms of zone of inhibition (in millimeters) in well diffusion assay, minimum inhibitory concentration (MIC) and minimum fungicidal concentration (MFC). All EOs were active against the tested microorganisms and the diameter of the halos varied from 7 to 26 mm (Table 2). The EO of *D. echinophora* showed the higherst inhibition power (16–26 mm).

Minimal inhibitory concentration (MIC) and minimum fungicidal concentration (MFC) are listed in Table 3. The results indicated that the EOs of *D. echinophora*, *X. emarginata* and *X. frutescens* showed a high ability to inhibit the microorganism *C. famata* and exhibited MIC values of 0.07, 0.019 and 0.62 $\mu L.mL^{-1}$, respectively. Based on MFC results, the EOs of *D. echinophora*, *X. frutescens* and *X. emarginata* displayed a fungicide activity against *C. tropicalis* (2.5 $\mu L.mL^{-1}$), *C. krusei* (2.5 $\mu L.mL^{-1}$), and *C. auris* (5 $\mu L.mL^{-1}$), respectively. The results presented by the different fractions of essential oils in the present study may be associated with the presence of different classes of compounds such as hydrocarbon monoterpenes, oxygenated monoterpenes, hydrocarbon sesquiterpenes, and oxygenated sesquiterpenes, as several literatures have shown that these classes have potential antifungal activities [53–56], being more specific the major compounds identified in essential oils rich in β-phellandrene, cryptone, spathulenol, and β-pinene have shown potential antifungal activities [57–60], furthermore, recent studies on fungal infections reported that the pathogens *C. albicans*, *C. tropicalis*, *C.*

**Table 2. In vitro effect of the essential oils of *Annonaceae* species on medically important yeasts using agar disc diffusion method.**

| Microorganism | Halo Diameter (mm) | | | | |
|---|---|---|---|---|---|
| | $A_{dol}$ | $D_{ech}$ | $G_{sca}$ | $X_{ema}$ | $X_{fru}$ |
| *C. albicans* | 10 | 16 | 8 | 8 | 10 |
| *C. auris* | 7 | 20 | 7 | 10 | 9 |
| *C. famata* | 9 | 26 | 8 | 12 | 12 |
| *C. krusei* | 9 | 19 | 8 | 8 | 10 |
| *C. tropicalis* | 9 | 19 | - | 10 | 9 |

$D_{ech}$ = *Duguetia echinophora*; $X_{ema}$ = *Xylopia emarginata*; $X_{fru}$ = *Xylopia frutescens*. Experiments performed in duplicate.

**Table 3. Minimal inhibitory concentration (MIC) and minimal fungicidal concentration (MFC) of essential oils of *Annonaceae* species on medically important yeasts; data expressed in µL.mL$^{-1}$.**

| Microorganism | D$_{ech}$ | | X$_{ema}$ | | X$_{fru}$ | |
|---|---|---|---|---|---|---|
| | MIC | MFC | MIC | MFC | MIC | MFC |
| *C. albicans* | 0.31 | > 5 | 2.5 | > 5 | 5 | >5 |
| *C. auris* | 0.62 | > 2.5 | 5 | 5 | 5 | >5 |
| *C. famata* | 0.07 | > 0.31 | 0.019 | > 0.07 | 0.62 | > 2.5 |
| *C. krusei* | 0.31 | > 1.25 | 0.62 | > 2.5 | 2.5 | 5 |
| *C. tropicalis* | 0.62 | 2.5 | 2.5 | > 5 | 5 | > 5 |

D$_{ech}$ = *Duguetia echinophora*; X$_{ema}$ = *Xylopia emarginata*; X$_{fru}$ = *Xylopia frutescens*. Experiments performed in duplicate.

*krusei*, and *C. auris* are related to bloodstream infections, known as invasive candidiasis, and commonly associated with high mortality rates due to the loss of efficiency of antifungal treatments caused by the emergence of resistant forms of pathogens [61]. Alternative treatments capable of controlling such fungal infections may include the EOs which are a source of substances that, isolated or combined, can inhibit or control fungal growth.

Previous studies demonstrated the antifungal activity of the EOs from some *Annonaceae* species against *Candida* spp., however, there is no specific study of the antifungal activity of EOs from *A. dolichocarpa*, *D. echinophora*, *G. scandens*, *X. emarginata* and *X. frutescens*. The EO from the leaves of *A. brevipes*, composed mainly of β-eudesmol (13.16%), α-eudesmol (13.05%), γ-eudesmol (7.54%), and guaiol (5.12%), showed antifungal inhibitory effect against *C. albicans* and *C. parapsilosis* (MIC values of 50.0 to 100.0 µg.mL$^{-1}$, respectively) [12]. The EO from the leaves of *A. vepretorum*, rich in bicyclogermacrene (43.7%), spathulenol (11.4%), α-phelandrene (10.0%), α-pinene (7.1%), (*E*)-β-ocimene (6.8%), germacrene D (5.8%) and *p*-cymene (4.2%), exhibited antimicrobial activity against *C. albicans* and *C. tropicalis*, with MIC values the 5000 and 100 µg.mL$^{-1}$, respectively [13]. *D. lanceolata* EO composed mainly of β-elemene (12.7%), caryophyllene oxide (12.4%) and β-selinene (8.4%), inhibited the growth of *C. albicans* (MIC values 60.0 µg.mL$^{-1}$) [14].

The results found in the present paper indicated that those EOs tested could be a source of bioactive substances with antifungal activity, since terpenes and their derivatives was the class of compounds identified in the highest proportion. These compounds exhibit several pharmacological activities, including antimicrobial activity, and are part of the mechanisms of oxidative phosphorylation and oxygen uptake, which are extremely important for microbial survival, the interaction between terpene and microorganism causes alteration in cellular respiration resulting in the uncoupling of oxidative phosphorylation [9]. However the mode of action must be better understand once that the tested EOs are a complex mix of mono and sesquiterpenes, which can act synergistically or antagonistically.

## 3.3. Molecular docking

Computer-aided drug design has supported the investigation of new drugs against various diseases worldwide. This paper mainly used molecular modeling approaches to investigate how major compounds of essential oils from Annonaceae species can interact with molecular targets of *Candida* sp. The compounds used in the essential oil were the majority compounds; we consider that the majority of compounds have a concentration greater than 5% of the oil composition [62–64].

Furthermore, to communicate our results *in silico* and what was observed experimentally in the treatment of candidiasis, we selected molecular targets that are the target of commercial

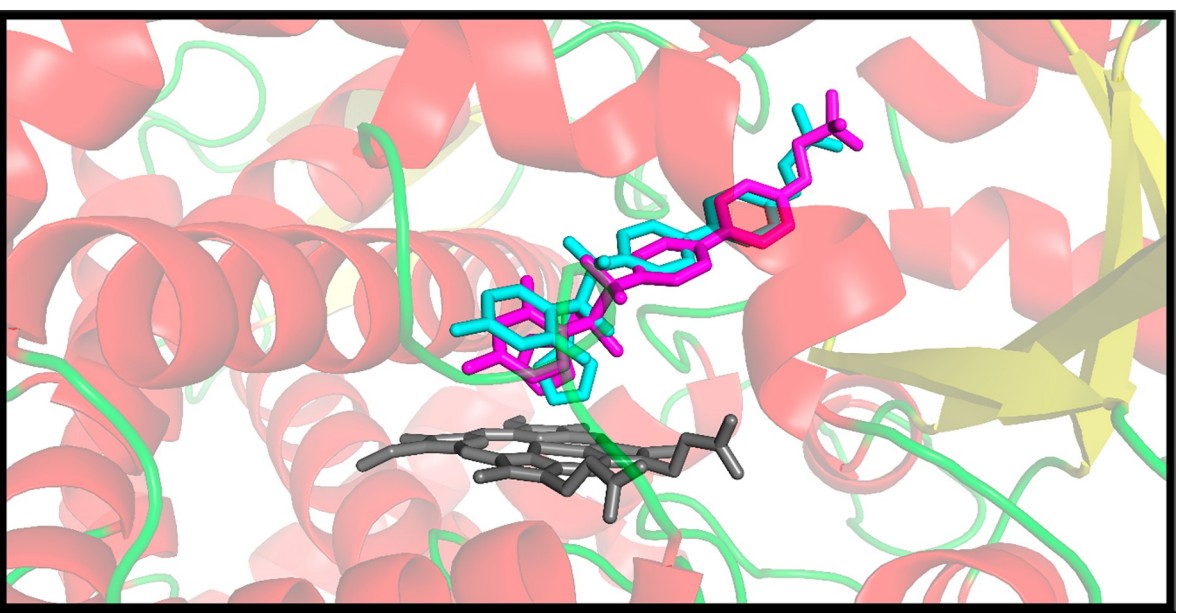

**Fig 1. The structure was obtained by redocking (magenta), overlapping the crystallographic structure (cyan) of CYP51 complex.**

drugs for this disease. We used the proteins Sterol 14α-Demethylase Cytochrome P450 as molecular targets because they are targets of commercial drugs such as triazoles that inhibit ergosterol biosynthesis (fluconazole, itraconazole, voriconazole, posaconazole) [65–67].

Before proceeding with molecular docking, we first evaluated whether the docking protocol can reproduce *in silico* the binding mode of interaction of the crystallographic ligand. For this purpose, each crystallographic ligand was redocked using the template docking feature implemented in the Molegro Virtual Docker 5.5 program. The fitness evaluation redocked pose was evaluated by considering the RMSD values and docking scores. According to the literature, a docking protocol is suitable for thorough investigations when the RMSD between the crystallographic ligand and the redocked ligand has a value equal to or less than 2 angstroms [68–70]. The RMSD value between the crystallographic and redocked ligand showed a slight deviation reaching a value of 1.38 Å (Fig 1). Therefore, our docking protocol demonstrated to reproduce an experimentally obtained binding mode.

Then the β-phellandrene, cryptone, spathulenol, and pinene compounds were docked in the sterol 14α-demethylase cytochrome P450 binding pocket, and the chemical interactions responsible for the interaction with the active site can be seen in Fig 2.

The compound β-phellandrene established hydrophobic interactions with residues of Leu376, His 377, Phe 233, Leu121, Met508, Pro230 and Val510, while van der Waals interactions were formed with Val509, Tyr505, Ser507, Ser506, Phe380, Tyr118, Ser378 and Pro375. Cryptone was also able to interact with residues of Met508, Tyr118, Phe380, Phe233, and Leu 121, while interactions with Ile379, Leu376, Pro375, Val509, Ser378, His377, and Tyr64 were all van der Waals interactions. Spathulenol formed more van der Waals interactions than hydrophobic interactions. Van der Waals interactions were formed with Thr122, Met508, Phe126, Gly307, His310 and Thr311; the hydrophobic interactions were with Tyr118, Leu121, Phe233, Leu376, Tyr132 and Phe228. Pinene interacted with residues of Mey508, Pro230, Leu121, Phe233, His377, and Leu376 through hydrophobic interactions, and van der Waals interactions were formed with Phe380, Tyr64, Ser378, Val509, Pro375, Val510, and Ser507.

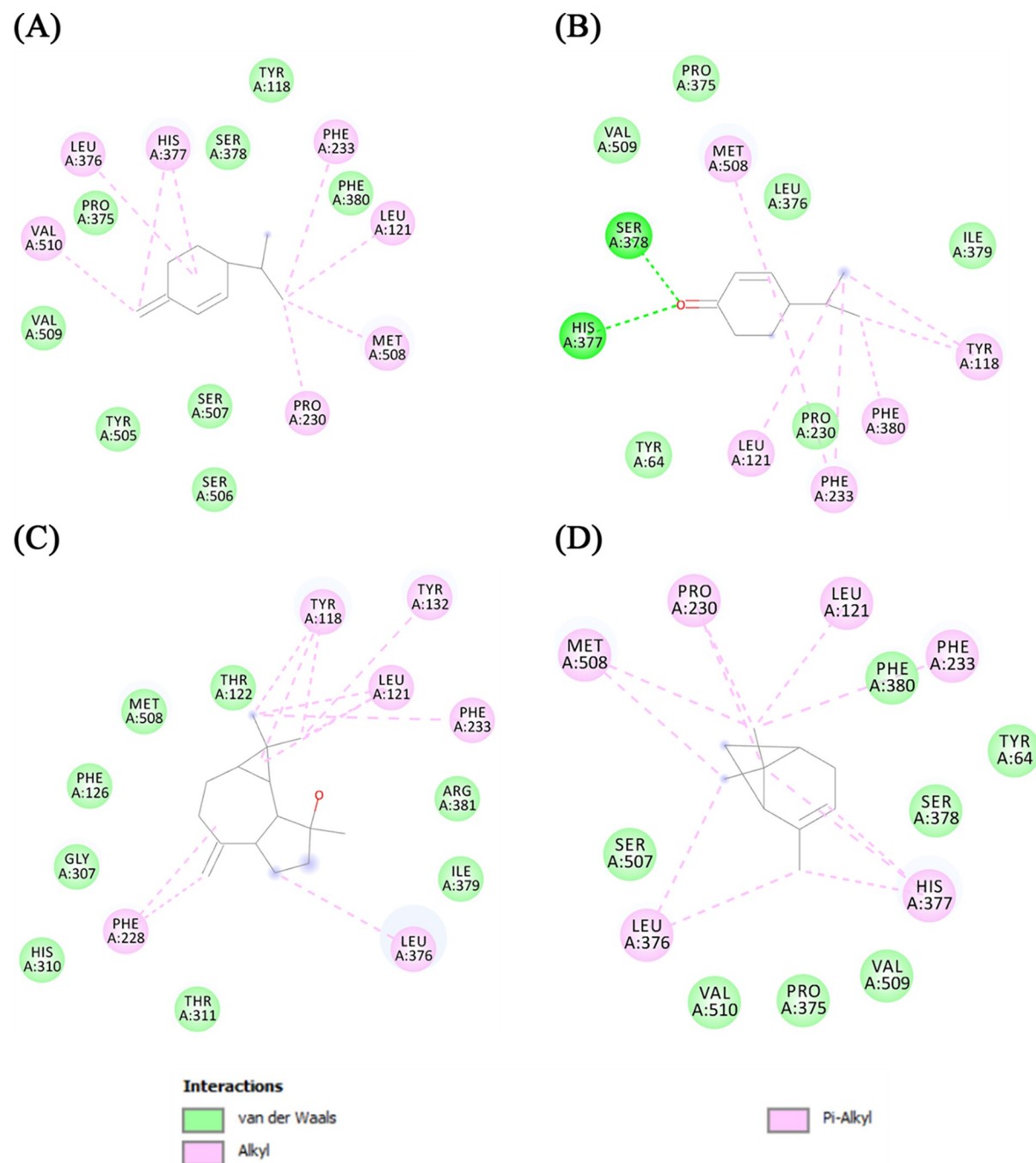

**Fig 2.** Representation of the interactions established with the binding pocket of Sterol 14α-Demethylase Cytochrome P450 with the compounds (A) β-Phellandrene, (B) Cryptone, (C) Spathulenol and (D) β-Pinene.

## 4. Conclusions

This study reports the chemical composition and antifungal activity of five *Annonaceae* EOs against standard strains associated with human infections, namely *C. albicans*, *C. auris*, *C. famata*, *C. krusei* and *C. tropicalis*. The EO yield varied from 0.16 to 1.53% and the GC/MS analysis of the EOs showed that monoterpenes were predominant in the EOs of *D. echinophora*

(47.24%), *G. scandens* (57.15%) and X. *frutescens* (89.18%), while sesquiterpenes characterized the EOs of *A. dolichocarpa* (51.15%) and *X. emarginata* (57.20%). The chemical composition found in this work is in agreement with other species of *Annonaceae*, characterized by mono and sesquiterpenes. All EOs were active against the tested strains of *Candida*. The EOs of *D. echinophora*, *X. emarginata* and *X. frutescens* showed a high ability to inhibit the microorganism *C. famata* (MIC values of 0.07, 0.019 and 0.62 $\mu L.mL^{-1}$, respectively). The results of MFC showed that the EOs of *D. echinophora*, *X. frutescens* and *X. emarginata* were found to display a fungicide activity against the species *C. tropicalis* (2.5 $\mu L.mL^{-1}$), *C. krusei* (2.5 $\mu L.mL^{-1}$) and *C. auris* (5 $\mu L.mL^{-1}$), respectively. The presence of compounds belonging to the class of the terpenes in the EOs may explain the antifungal action of the tested EOs through synergistic or antagonistic effects, however, there is a need to carry out *in vivo* clinical trials focusing on the antimicrobial activity of theses EOs. The molecular docking study suggested that the major compounds of the EOs of Annonaceae species can interact with molecular targets of *Candida* spp.

## Acknowledgments

The author M.M.C. and M.S.O thanks CAPES for the Ph.D, and Postdoctoral scholarship, respectively. The authors thanks Edital—PAPQ/PROPESP—Universidade Federal do Para.

## Author Contributions

**Conceptualization:** Márcia Moraes Cascaes.

**Formal analysis:** Márcia Moraes Cascaes, Silvia Helena Marques da Silva, Jorddy Neves Cruz, Ângelo Antônio Barbosa de Moraes, Lidiane Diniz do Nascimento, Oberdan Oliveira Ferreira.

**Supervision:** Mozaniel Santana de Oliveira, Giselle Maria Skelding Pinheiro Guilhon, Eloisa Helena de Aguiar Andrade.

**Validation:** Giselle Maria Skelding Pinheiro Guilhon, Eloisa Helena de Aguiar Andrade.

**Visualization:** Mozaniel Santana de Oliveira, Giselle Maria Skelding Pinheiro Guilhon, Eloisa Helena de Aguiar Andrade.

**Writing – original draft:** Márcia Moraes Cascaes.

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
