## [Decision Letter · Decision Letter 0]

25 Jun 2023

PONE-D-23-15730Exploring the Chemical Composition, In Vitro and in Silico Study of the Antimicrobial Properties of Annonaceae Species Essential Oils  from the AmazonPLOS ONE

Dear Dr. de Oliveira,

Thank you for submitting your manuscript to PLOS ONE. After careful consideration, we feel that it has merit but does not fully meet PLOS ONE’s publication criteria as it currently stands. Therefore, we invite you to submit a revised version of the manuscript that addresses the points raised during the review process.

Please consider and answer the questions by both reviewers

We look forward to receiving your revised manuscript.

Kind regards,

Guadalupe Virginia Nevárez-Moorillón, Ph.D.

Academic Editor

PLOS ONE

Journal Requirements:

4. We noticed you have some minor occurrence of overlapping text with the following previous publication(s), which needs to be addressed:

Insight into the Interaction Mechanism of Nicotine, NNK, and NNN with Cytochrome P450 2A13 Based on Molecular Dynamics Simulation - https://doi.org/10.1021/acs.jcim.9b00741

In your revision ensure you cite all your sources (including your own works), and quote or rephrase any duplicated text outside the methods section. Further consideration is dependent on these concerns being addressed.

Reviewers' comments:

Reviewer's Responses to Questions

**Comments to the Author**

1. Is the manuscript technically sound, and do the data support the conclusions?

Reviewer #1: Yes

Reviewer #2: Partly

2. Has the statistical analysis been performed appropriately and rigorously? 

Reviewer #1: Yes

Reviewer #2: No

3. Have the authors made all data underlying the findings in their manuscript fully available?

Reviewer #1: Yes

Reviewer #2: Yes

4. Is the manuscript presented in an intelligible fashion and written in standard English?

Reviewer #1: Yes

Reviewer #2: No

5. Review Comments to the Author

Reviewer #1: Some concerns:

Why cytotoxic activity and selectivity index were not done or were not presented?

Title

The term antimicrobial is too broad in this title as only Candida strains were used. I suggest changing from 'antimicrobial properties' for ‘anticandidal properties’.

Abstract

- In the abstract section it is not recommended to use acronyms such as GC-MS, MIC, MFC.

- All strain names must be spelled out

-Specify if all essential oil present same qualitative composition. What about quantitative?

Materials and Methods section

2.4. Antifungal Activity of the Essential Oils

Include program used to calculate the MIC values.

2.5. Molecular Docking

Line 160: using Gaussian09 program

Line 168: ‘energy threshold equal to 100’ ??? What’s the unity?

References section

The authors must standardize the references according to the rules of the PlosONE Journal, i.e., number 07, 17,23,27, 30, 34, 36, 39, 40, 41, 43 and 52.

Reviewer #2: The manuscript PONE-D-23-15730 entitled “Exploring the Chemical Composition, In Vitro and in Silico Study of the Antimicrobial Properties of Annonaceae Species Essential Oils from the Amazon” reports the composition of the essential oils (EOs) of several plants of the Annonaceae family and explored their potential interaction with the proteins Sterol 14α-Demethylase Cytochrome P450, as their molecular target using a molecular docking approach. The Knowledge added by the manuscript is qualified in the context of the antimicrobial activity of the tested EOs, but just include the molecular docking is not sufficient to improve the value of the study, more validation and exploitation on the use of these EOs is required to considerer this manuscript to be published.

Following the suggestion of minor alterations

1. There is a link in the title to an article ( https://doi.org/10.3390/molecules17089540)

2. Line 36 MFC first in full

3. Line 36-37 rewrite the sentence is Grammarly confused

4. Line 41 change ntimicrobial to antimicrobial

5. Line 82 change were use to were used

6. Line 92 indicate the first and surname of J. O.

7. Line 110 revise the order of your references here and all over the manuscript

8. Line 133 indicate the volume used and the respective concentration for nystatin.

9. Line 146 eliminate in a bacteriological incubator

10. Line 148 and this concentration was considered the Minimum Inhibitory Concentration?

11. Line 152 MIC in full first

12. Line 153 what dilution? what the authors diluted?

13. Line 155 this is not clear. Rewrite this section. Moreover, the cited reference is in Portuguese.

14. Line 234 check the sentence

15. Line 272 As the authors can observe from the literature cited, is recommended to use the concentration of the EO in ug/mL or mg/mL, and not v/v.

16. Line 272-278 Line 266-278. A proper discussion must be provided, not a statement of the results reported by other studies.

17. Line 285 what paper? the current study or the previous mentioned?

18. Figure 1 and 2 The legend of the Figures must be at the bottom of the Figure.

19. Line 385 the format of the references requires a deep revision.

6. PLOS authors have the option to publish the peer review history of their article (what does this mean?). If published, this will include your full peer review and any attached files.

Reviewer #1: **Yes: **FRANCISCO JOSE TORRES DE AQUINO

Reviewer #2: No

---

## [Author Response · Author response to Decision Letter 0]

19 Jul 2023

Response to Reviewers comments:

Reviewer's Answers to Questions

Comments to the Author

1. Is the manuscript technically sound, and do the data support the conclusions?

Reviewer #1: Yes

Reviewer #2: Partly

2. Has the statistical analysis been performed appropriately and rigorously?

Reviewer #1: Yes

Reviewer #2: No

3. Have the authors made all data underlying the findings in their manuscript fully available?

Reviewer #1: Yes

Reviewer #2: Yes

4. Is the manuscript presented in an intelligible fashion and written in standard English?

Reviewer #1: Yes

Reviewer #2: No

5. Review Comments to the Author

Reviewer #1:

Some concerns:

Why cytotoxic activity and selectivity index were not done or were not presented?

Answer: Unfortunately, our laboratory does not have the necessary resources (funders) to carry out the tests suggested by you, we are sorry for that.

Title

The term antimicrobial is too broad in this title as only Candida strains were used. I suggest changing from 'antimicrobial properties' to 'anticandidal properties'.

Answer: We changed the title as suggested by you, thank you very much.

“Exploring the Chemical Composition, In Vitro and in Silico Study of the Anticandidal Properties of Annonaceae Species Essential Oils from the Amazon” 

Abstract

- In the abstract section it is not recommended to use acronyms such as GC-MS, MIC, MFC.

Answer: This section has been corrected as per your recommendation.

- All strain names must be spelled out

Answer: We carry out the correction.

-Specify if all essential oil present same qualitative composition. What about quantity?

Answer: each plant species studied presented a concentration of essential oil, consequently the chemical composition in qualitative and quantitative terms varied, this can be observed more clearly in section 3. Results and Discussions 3.1 Chemical Composition and Yield of the Essential Oils, Table 1, thank you for your inquiry.

New Abstract: Chemical composition of the essential oils (EOs) from the leaves of five Annonaceae species found in the amazon region was analyzed by Gas chromatography coupled to mass spectrometry. The antifungal activity of theses EOs was tested against Candida albicans, Candida auris, Candida famata, Candida krusei and Candida tropicalis. In addition, an in silico study of the molecular interactions was performed using molecular modeling approaches. Spathulenol (29.88%), α-pinene (15.73%), germacra-4(15),5,10(14)-trien-1-α-ol (6.65%), and caryophylene oxide (6.28%) where the major constitents from the EO of Anaxagorea dolichocarpa. The EO of Duguetia echinophora was characterized by β-phellanderene (24.55%), cryptone (12.43%), spathulenol (12.30%), and sabinene (7.54%). The major compounds of the EO of Guatteria scandens where β-pinene (46.71%), α-pinene (9.14%), bicyclogermacrene (9.33%), and E-caryophyllene (8.98%). The EO of Xylopia frutescens was characterized by α-pinene (40.12%) and β-pinene (36.46%). Spathulenol (13.8%), allo-aromadendrene epoxide (8.99%), thujopsan-2-α-ol (7.74%), and muurola-4,10(14)-dien-1-β-ol (7.14%) were the main chemical constituents reported in Xylopia emarginata EO. All EOs were active against the strains tested and the lowest inhibitory concentrations were observed for the EOs of D. echinophora, X. emarginata, and X. frutescens against C. famata the Minimum Inhibitory Concentration values of 0.07, 0.019 and 0.62 µL.mL-1, respectively. The fungicidal action was based on results of minimum fungicidal concentration and showed that the EOs showed fungicide activity against C. tropicalis (2.5 µL.mL-1), C. krusei (2.5 µL.mL-1) and C. auris (5 µL.mL-1), respectively. The computer simulation results indicated that the major compounds of the EOs can interact with molecular targets of Candida spp.

Materials and Methods section

2.4. Antifungal Activity of the Essential Oils

Include program used to calculate the MIC values.

Answer: To calculate the minimum inhibitory concentrations (MIC), the protocols of Kowalska-Krochmal et al., [1] and minimum fungicidal concentration (MFC) Rex et al., [2], Both using Excel software.

1. Kowalska-Krochmal B, Dudek-Wicher R. The Minimum Inhibitory Concentration of Antibiotics: Methods, Interpretation, Clinical Relevance. Pathogens. 2021;10:165. doi:10.3390/pathogens10020165

2. Rex JH, Alexander BD, Andes D, Arthington-Skaggs B, Brown SD, Chaturvedi V, et al. Reference method for broth dilution antifungal susceptibility testing of yeasts. Approved standard, 3rd ed. Clin Lab Stand Inst. 2008;28:0–13. Available: https://clsi.org/media/1461/m27a3_sample.pdf

2.5. Molecular Docking

Line 160: using Gaussian09 program

Answer: We made the change as recommended, thank you very much.

Line 168: 'energy threshold equal to 100' ??? What's the unity?

Answer: The “energy threshold” has no unit of measurement.

References section

The authors must standardize the references according to the rules of the PlosONE Journal, i.e., number 07, 17,23,27, 30, 34, 36, 39, 40, 41, 43 and 52.

Answer: The references have been changed and corrected with the help of the Mendeley App using the style provided by Plos One.

Reviewer #2: 

The manuscript PONE-D-23-15730 entitled “Exploring the Chemical Composition, In Vitro and in Silico Study of the Antimicrobial Properties of Annonaceae Species Essential Oils from the Amazon” reports the composition of the essential oils (EOs) of several plants of the Annonaceae family and explored their potential interaction with the proteins Sterol 14α-Demethylase Cytochrome P450, as their molecular target using a molecular docking approach. The Knowledge added by the manuscript is qualified in the context of the antimicrobial activity of the tested EOs, but just include the molecular docking is not sufficient to improve the value of the study, more validation and exploitation on the use of these EOs is required to considerer this manuscript to be published.

Following the suggestion of minor alterations

1. There is a link in the title to an article ( https://doi.org/10.3390/molecules17089540)

Answer: The link in the title was removed “the new tile: Exploring the Chemical Composition, In Vitro and in Silico Study of the Anticandidal Properties of Annonaceae Species Essential Oils from the Amazon” 

2. Line 36 MFC first in full

Answer: This sentence was checked

3. Line 36-37 rewrite the sentence is Grammarly confused

Answer: This sentence was checked

4. Line 41 change ntimicrobial to antimicrobial

Answer: This sentence was checked

5. Line 82 change were use to were used

Answer: This sentence was checked

6. Line 92 indicate the first and surname of J. O.

Answer: The first and surname of J. O were added

7. Line 110 revise the order of your references here and all over the manuscript

Answer: References have been reviewed.

8. Line 133 indicate the volume used and the respective concentration for nystatin.

Answer: paper discs were impregnated with a 30 μL nystatin solution.

9. Line 146 eliminate in a bacteriological incubator

Answer: This sentence was checked

10. Line 148 and this concentration was considered the Minimum Inhibitory

Concentration?

Answer: Yes, as it was according to our previously published study Ferreira OO, da Silva SHM, de Oliveira MS, Andrade EH de A. Chemical Composition and Antifungal Activity of Myrcia multiflora and Eugenia florida Essential Oils. Molecules. 2021;26: 7259. doi:10.3390/molecules26237259

11. Line 152 MIC in full first

Answer: This sentence was checked

12. Line 153 what dilution? what the authors diluted?

Answer: DMSO was used for the dilutions.

13. Line 155 this is not clear. Rewrite this section. Moreover, the cited reference is in

Portuguese.

Answer: We revised the sentence and changed the reference to a universal

language

14. Line 234 check the sentence

Answer: This sentence was checked

15. Line 272 As the authors can observe from the literature cited, is recommended to use the concentration of the EO in ug/mL or mg/mL, and not v/v.

Answer: The results were adjusted and expressed in μL.mL-1

16. Line 272-278 Line 266-278. A proper discussion must be provided, not a statement

of the results reported by other studies.

Answer: We have improved the discussion of this section, thank you.

17. Line 285 what paper? the current study or the previous mentioned?

Answer: We refer to our paper. The paragraph has been rewritten.

18. Figure 1 and 2 The legend of the Figures must be at the bottom of the Figure.

Answer: These legends were added the bottom of the Figures.

19. Line 385 the format of the references requires a deep revision.

Answer: The references were revised.

On behalf of all authors I would like to thank the reviewers and editor for their valuable recommendations, we believe that all inquiries were instrumental in improving the quality of the manuscript.

Sincerely,

Prof. Dr. Mozaniel de Oliveira 

Museu Pareaense Emílio Goeldi

---

## [Editor Report · Decision Letter 1]

31 Jul 2023

Exploring the Chemical Composition, In Vitro and in Silico Study of the Anticandidal Properties of Annonaceae Species Essential Oils  from the Amazon

PONE-D-23-15730R1

Dear Dr. de Oliveira,

We’re pleased to inform you that your manuscript has been judged scientifically suitable for publication and will be formally accepted for publication once it meets all outstanding technical requirements.

Kind regards,

Guadalupe Virginia Nevárez-Moorillón, Ph.D.

Academic Editor

PLOS ONE
---

## [Editor Report · Acceptance letter]

15 Aug 2023

PONE-D-23-15730R1 

 Exploring the Chemical Composition, In Vitro and in Silico Study of the Anticandidal Properties of Annonaceae Species Essential Oils  from the Amazon 

Dear Dr. de Oliveira:

I'm pleased to inform you that your manuscript has been deemed suitable for publication in PLOS ONE. Congratulations! Your manuscript is now with our production department. 

Kind regards, 

on behalf of

Dr. Guadalupe Virginia Nevárez-Moorillón 

Academic Editor

PLOS ONE